# Artifact-Free Evaluation of Choriocapillaris Perfusion in Central Serous Chorioretinopathy

**Maria A. Burnasheva, Alexei N. Kulikov and Dmitrii S. Maltsev ***

Department of Ophthalmology, Military Medical Academy, 194044 St. Petersburg, Russia;
maria.andreevna1@gmail.com (M.A.B.); alexey.kulikov@mail.ru (A.N.K.)
* Correspondence: glaz.med@yandex.ru; Tel.: +7-(905)273-05-82

**Abstract:** In this study, using optical coherence tomography angiography (OCTA) we evaluated choriocapillaris perfusion in eyes with central serous chorioretinopathy (CSC) after excluding all possible artifacts caused by alterations of neurosensory retina or retinal pigment epithelium (RPE). We have included 22 unilateral acute CSC patients (18 males and four females, $41.8 \pm 5.7$ years) and 18 healthy subjects (13 males and five females, $40.9 \pm 9.7$ years). The number of flow voids per 1 mm$^2$ of scan area and percentage of flow signal area was calculated based on choriocapillaris slab of $3 \times 3$ mm$^2$ OCTA scans after excluding all possible artifacts caused by alterations of neurosensory retina or RPE. The percentage of flow signal area after the exclusion of neurosensory retina and RPE alterations in the eyes of healthy individuals was higher than in CSC eyes ($p = 0.006$) and fellow unaffected eyes of CSC patients ($p = 0.02$). The number of flow voids larger than 25,000 μm$^2$ in eyes of healthy individuals was lower than in the CSC eyes ($p = 0.0006$). There were no statistically significant differences in study parameters between CSC eyes and fellow eyes of CSC patients ($p > 0.05$). The general decrease of choriocapillaris perfusion in both eyes of CSC patients exists independently of the presence of acute disease or asymptomatic structural RPE changes.

**Keywords:** central serous chorioretinopathy; choriocapillaris; choroid; optical coherence tomography angiography

## 1. Introduction

The evaluation of choriocapillaris circulation sheds light on the pathophysiology of central serous chorioretinopathy (CSC). The first mention of choroidal ischemia and related retinal pigment epithelium (RPE) alteration in CSC was by Hayashi [1] and was based on indocyanine green angiography. However, optical coherence tomography angiography (OCTA) has allowed higher resolution in the evaluation of choriocapillaris. Using OCTA, many authors have described an increase of the total area of flow signal voids in choriocapillaris in CSC [2–5] and have shown colocalization of flow signal voids and underlying choroidal pachyvessels [2,4,5].

However, the interpretation of choriocapillaris circulation with OCTA requires close attention. OCTA images may contain false information on hypo- or hyperperfusion due to the number of artifacts. Several authors have reported attenuation of the flow signal in the choriocapillaris under the detached neurosensory retina, RPE detachments, and in cases of irregularity or hyperreflectivity of outer retina [6,7]. All of this suggests that structural changes in overlying neurosensory retina and RPE play an important role in the imaging of blood flow in the choriocapillaris with OCTA. Although most CSC studies included only eyes without neurosensory detachment, the impact of RPE alterations was not taken into consideration. However, avoiding RPE-related artifacts are important since these changes may persist, even after resolution of subretinal fluid (SRF) in acute CSC. RPE-related artifacts may be particularly important for the analysis of fellow eyes of CSC patients where reduction of choriocapillaris perfusion is not as significant as in eyes with

acute CSC. The artifactual changes of perfusion may have more impact on the results of choriocapillaris imaging in these eyes. In a recent study of Cakir in patients with acute CSC, the increased OCTA signal in the choriocapillaris was shown over the areas with atrophic RPE changes, while the attenuated OCTA signal was found mostly beneath the areas of accumulation of SRF [8]. However, until now it has not been established if the flow deficit in the choriocapillaris beneath RPE detachments and other RPE abnormalities is not artifactual. In this study we, therefore, evaluated the status of choriocapillaris perfusion with OCTA in eyes of CSC patients excluding all possible artifacts related to the neurosensory retina or RPE alterations.

## 2. Materials and Methods

In this retrospective study we have included two groups of subjects: patients with unilateral acute CSC and healthy volunteers. Inclusion criteria for CSC patients were aged 25–55 years, resolved acute CSC or acute CSC with neurosensory detachment occupying less than 1/6 of the scan area. Inclusion criteria for healthy volunteers were aged 25–55 years, negative smoking status, absence of any known comorbidity. Exclusion criteria for CSC patients was bilateral acute CSC (defined as the presence of neurosensory detachment) or known history of acute CSC in the fellow eye. Exclusion criteria for both groups were myopia more than 6.0 D or hyperopia more than 2.0 D, any retinal diseases (excluding CSC for the groups of CSC patients), diabetes mellitus, OCTA scan quality 8/10 and lower, and the presence of motion artifacts.

All patients received a comprehensive ophthalmic examination and OCTA examination with RTVue-XR Avanti (Optovue, Fremont, CA, USA, software version 2017.1.0.150). Images of default choriocapillaris slab of $3 \times 3$ mm$^2$ OCTA scans (consisted of 304 doubled B-scans each of 304 A-scans, $600 \times 600$ pixels) were used for analysis. Only one eye of each participant was included in the healthy group. Both eyes of CSC patients were included in two subgroups: (1) CSC eyes (resolved acute CSC or acute CSC) and (2) fellow eyes.

The specific aim in evaluating the perfusion of choriocapillaris was the exclusion from the analysis of all areas of neurosensory retina or RPE alterations potentially interfering with OCT signal. To measure the choriocapillaris perfusion we used two independent approaches. With the first approach, we calculated the percentage of flow signal area after exclusion of regions of neurosensory retina or RPE alterations using OCT software. Firstly, for all images of the default choriocapillaris slab (between two Bruch's membrane segmentation lines with −9 and −31 μm offset), the total flow area was calculated. Next, the regions of neurosensory retina or RPE alterations, including all areas of accumulation of SRF, RPE detachments, irregularities of RPE or photoreceptors outer segment layer, hyperreflective subretinal material, were identified on corresponding structural en face images as hypo-/hyperreflective areas (Figure 1). The borders of these abnormal regions were confirmed by the revision of B-scans. To avoid possible inclusion of some abnormal borderline areas, only outer borders of abnormal regions have been delineated. Next, the total area of all abnormal regions and the total flow area within these areas were calculated using OCT software. For the remaining artifact-free area of the scan, the percentage of the flow area to total artifact-free area was calculated (Figure 2).

With the second approach, we evaluated the number of flow voids more than 10,000 μm$^2$ (400 pixels) and more than 25,000 μm$^2$ (1000 pixels) for the entire scan area excluding all abnormal regions in Fiji ((an expanded version of ImageJ version 1.51a (National Institutes of Health, Bethesda, MD, USA), available at fiji.sc free of charge). For this analysis, we have used the previously described autolocal thresholding algorithm of Phansalkar [9]. As described above, all abnormal regions were identified on structural en face images, manually traced, and measured. Additionally, using ROI-manager, the areas of choriocapillaris slab corresponding to abnormal regions were masked, and the remaining artifact-free area was subjected to flow voids calculation. The final measures were presented as the number of flow voids larger than 10,000 μm$^2$ and 25,000 μm$^2$ per 1 mm$^2$ of the artifact-free scan area (Figure 2). The cut-off value was chosen based on the known correlation between

voids larger than 10,000 $\mu m^2$ [10] and choriocapillaris damage. Since for larger voids (40,000 $\mu m^2$), an even higher correlation was revealed [11], we chose a second cut-off value of 25,000 $\mu m^2$.

MedCalc 18.4.1 (MedCalc Software, Ostend, Belgium) was used for statistical analysis. All data presented as mean ± standard deviation. The data were tested for the normality using the Kolmogorov–Smirnov test. The paired t-test was used to compare quantitative variables between two eyes of CSC patients. The independent t-test was used to compare eyes of CSC patients and eyes of healthy individuals. For the correction of multiple comparisons, $p < 0.025$ was considered as statistically significant based on Bonferroni's correction.

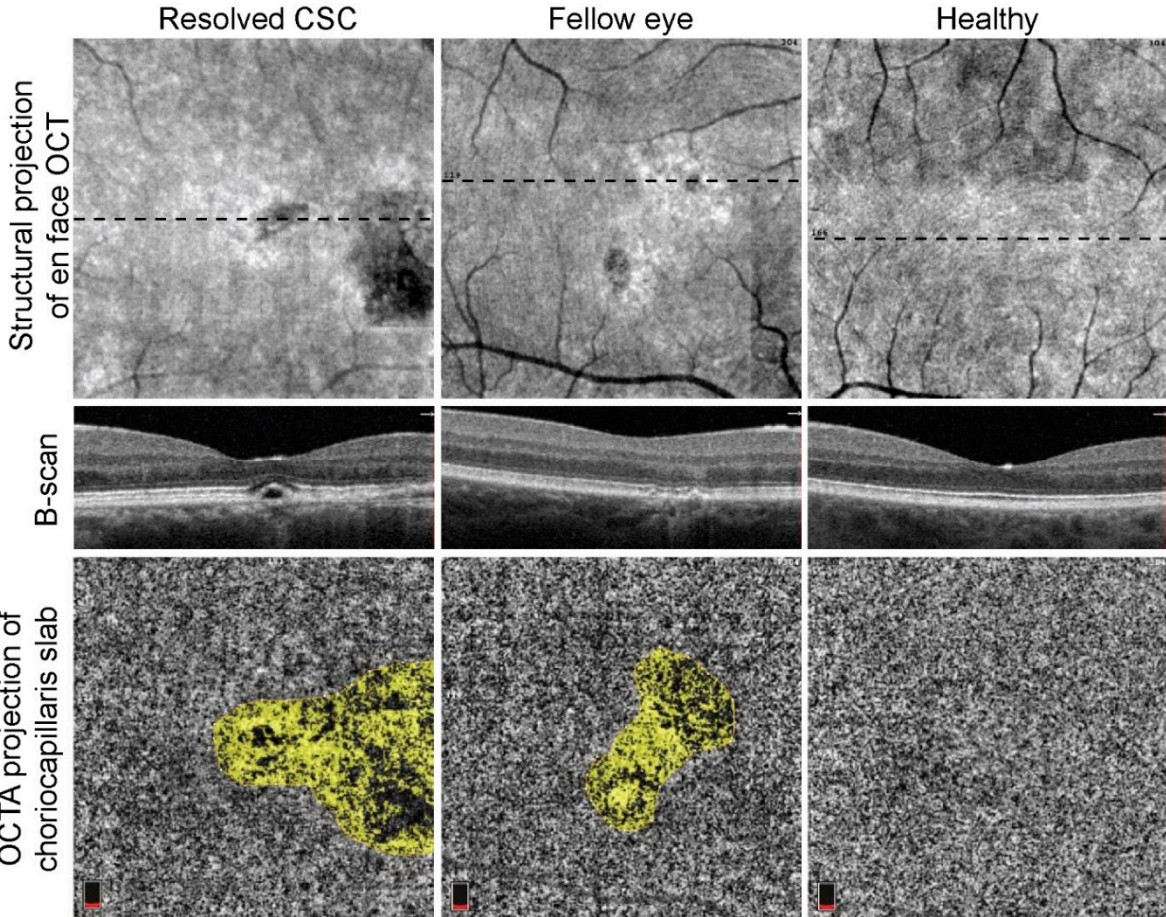

**Figure 1.** Representative examples of exclusion of the different artifacts in resolved central serous chorioretinopathy, fellow eye of central serous chorioretinopathy patient, and in eye of healthy person. CSC, central serous chorioretinopathy; OCT, optical coherence tomography; OCTA, optical coherence tomography angiography.

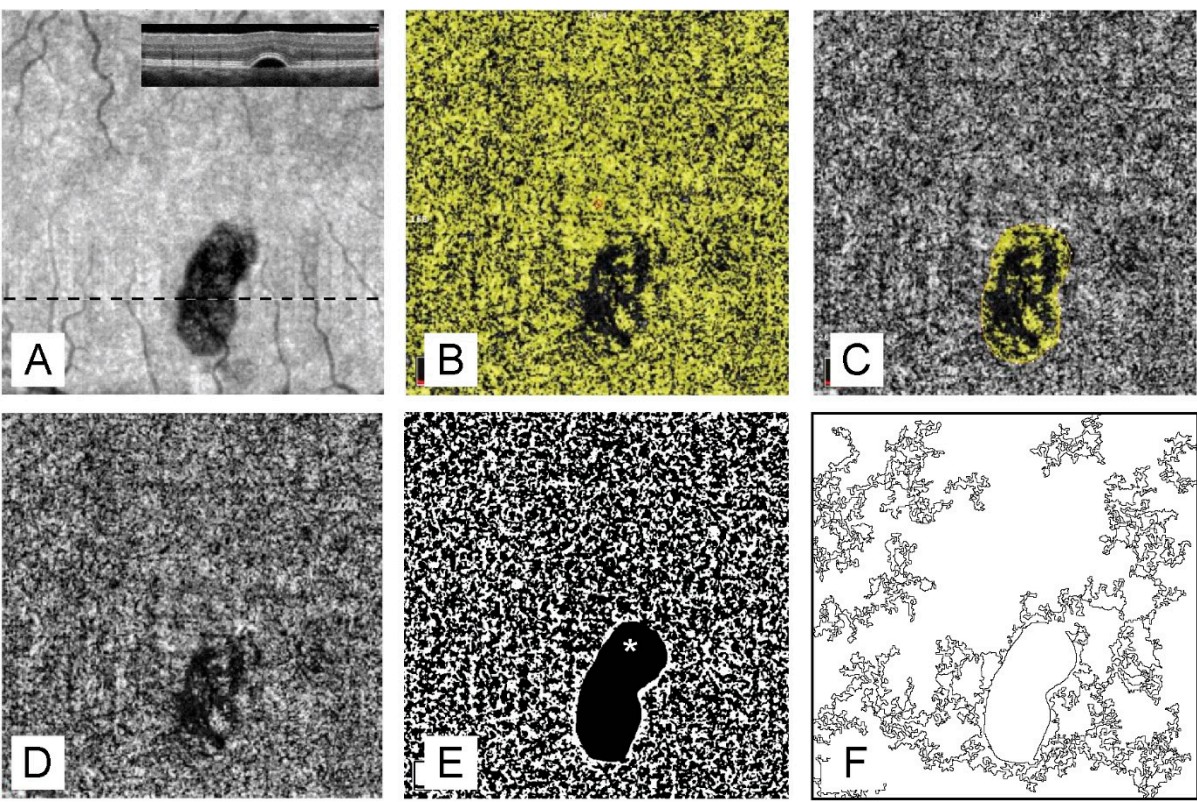

**Figure 2.** Evaluation of choriocapillaris perfusion after exclusion of the artifacts caused by retinal pigment epithelium alterations. (**A**). Structural en face projection of default choriocapillaris slab shows retinal pigment epithelium detachment. The dashed line represents the position of the cross-sectional scan in the inset. (**B**). Optical coherence tomography angiography (OCTA) projection of the default choriocapillaris slab showing flow area over the entire scan. (**C**). OCTA projection of the default choriocapillaris slab showing flow area under the RPE detachment. (**D**). OCTA projection of the default choriocapillaris slab shows a flow deficit under the RPE detachment. (**E**). Binarized image of the OCTA projection of the choriocapillaris slab after exclusion of the area under the RPE detachment (asterisk). (**F**). Distribution of flow voids larger than 25,000 $\mu m^2$.

## 3. Results

In this study we included 22 unilateral acute CSC patients (18 males and four females), with mean age of $41.8 \pm 5.7$ years. The group of healthy volunteers included 18 subjects (13 males and 5 females) with a mean age of $40.9 \pm 9.7$ years. There was no statistically significant difference in the male-to-female ratio and age between the groups ($p > 0.05$).

All OCTA images of CSC eyes required exclusion of neurosensory retinal detachment (2 eyes) or RPE alterations, while 5 (26.3%) fellow eyes demonstrating RPE alterations required exclusion before measurements.

The percentage of the flow signal area after exclusion of neurosensory retina and RPE alterations in the eyes of healthy subjects ($70.0 \pm 1.9\%$) was statistically significantly higher than that in CSC eyes ($68.0 \pm 3.1\%$, $p = 0.006$) or fellow eyes of CSC patients ($66.8 \pm 3.4\%$, $p = 0.02$) (Table 1).

The ratio of the number of flow voids larger than 10,000 $\mu m^2$ to the total area of scan after exclusion of neurosensory retina and RPE alterations in the eyes of healthy subjects ($7.59 \pm 1.37$ voids/mm$^2$) was not statistically significantly different than that in CSC eyes ($7.89 \pm 1.33$ voids/mm$^2$, $p = 0.51$) or fellow eyes of CSC patients ($8.10 \pm 1.20$ voids/mm$^2$, $p = 0.27$).

The ratio of the number of flow voids larger than 25000 $\mu m^2$ to the total area of scan after exclusion of neurosensory retina and RPE alterations in the eyes of healthy subjects ($1.32 \pm 0.68$ voids/mm$^2$) was not statistically significantly different than that in CSC eyes

$(2.31 \pm 0.92$ voids/mm$^2$, $p = 0.054$) but was statistically significantly higher in fellow eyes of CSC patients $(1.81 \pm 0.75$ voids/mm$^2$, $p = 0.0006$).

There were no statistically significant differences in study parameters between CSC eyes and fellow eyes of CSC patients ($p > 0.05$)

**Table 1.** Quantitative parameters of flow signal in the choriocapillaris in eyes of healthy subjects, affected and unaffected eyes of central serous chorioretinopathy patients.

| | CSC Patients | | Healthy Eye | *P*-Value | | |
|---|---|---|---|---|---|---|
| | **CSC Eye** | **Fellow Eye** | | **CSC Eye vs. Fellow Eye** | **CSC Eye vs. Healthy Eye** | **Fellow Eye vs. Healthy Eye** |
| **Voids (10,000 μm$^2$)/mm$^2$** | $7.89 \pm 1.33$ | $8.10 \pm 1.20$ | $7.59 \pm 1.37$ | 0.62 | 0.51 | 0.27 |
| **Voids (25,000 μm$^2$)/mm$^2$** | $2.31 \pm 0.92$ | $1.81 \pm 0.75$ | $1.32 \pm 0.68$ | 0.09 | 0.0006 | 0.054 |
| **Percentage of flow area, %** | $66.8 \pm 3.4$ | $68.0 \pm 3.1$ | $70.0 \pm 1.9$ | 0.28 | 0.006 | 0.02 |

CSC, central serous chorioretinopathy.

## 4. Discussion

The method for quantification of flow signal voids on OCTA images with Phansalkar auto local thresholding algorithm [9] was described by Spaide [10] and later used by a number of authors in similar studies [3,4,12]. These studies have shown an age-related increase in the number of flow voids in healthy subjects and its association with several risk factors, including hypertension, pseudodrusen, and age-related macular degeneration in the fellow eye.

Pachychoroidal spectrum is a specific point of interest for the study of changes of the choriocapillaris. Compared to healthy age-matched controls, the mean number of flow voids, the total flow voids area, and the mean area of flow voids were statistically significantly higher in eyes with pachychoroidal pigment epitheliopathy and CSC eyes [5,10]. The total flow void area increases with age, duration, and severity of CSC [3,5]. In addition, colocalization of large choriocapillaris flow voids with underlying pachyvessels and correlation between subfoveal choroidal thickness and presence of flow voids were found [4,5,12]. In fellow unaffected eyes of CSC patients, the total flow voids area and the mean area of flow voids were statistically significantly higher than in healthy subjects. After the resolution of SRF, the total flow voids area in acute CSC eyes was also higher than in fellow unaffected eyes. The total flow voids area in eyes with chronic/persistent CSC was higher than in eyes with acute CSC after the resolution of SRF [3]. However, the results of these studies do not allow comparison due to use of different devices, scan patterns, and calculation methods. Additionally, several authors note the possible distortion of choriocapillaris images by scanning artifacts or other factors, including neurosensory detachments, and RPE changes [4,7]. The main point of this study was therefore to exclude from analysis regions of the scan where choriocapillaris flow may be affected by overlaid structural changes of RPE or neurosensory retina.

Indeed, in the choriocapillaris slab, we observed areas of decreased flow signal under RPE irregularities and detachments. Although choriocapillaris hypoperfusion may be responsible for the alteration of overlaying RPE, structural changes of RPE inevitably interfere with the OCT scanning beam and reduce its intensity leading to a decrease of the flow signal. Since it is not possible to estimate the exact contribution of OCT signal attenuation to flow deficit under RPE alterations, we excluded these regions from the analysis. Two independent approaches were used to evaluate choriocapillaris perfusion, both of which took into account only artifact-free zones of the scan. Despite the exclusion of all zones potentially associated with artifactual change of the flow signal in the choriocapillaris, we found a statistically significant decrease in the flow signal area in eyes of CSC patients. This agrees with the increased number of flow voids larger than 25,000 μm$^2$ in eyes of CSC patients compared to the eyes of healthy subjects. However, this difference was statistically not significant for fellow unaffected eyes of CSC patients ($p = 0.054$). Additionally, paired eyes of CSC patients showed no difference at any parameter of choriocapillaris perfusion. All of this indicates an overall decrease of choriocapillaris perfusion in CSC patients, which

does not depend on the clinical manifestation of the disease or the presence of RPE alterations and neurosensory detachments. Taking into consideration previously reported differences between acute CSC and fellow unaffected eyes of CSC patients [3], we may conclude that this difference results from the higher prevalence of RPE and neurosensory retina changes in eyes with manifest CSC. Although, there is no clear evidence of the presence/absence of true deterioration of the choriocapillaris flow under RPE alterations and areas of SRF accumulation in eyes with CSC, we suggest that SRF is less likely to be associated with true flow deficit since the distribution of SRF depends on tertiary factors such as the location of the leak, the volume of the SRF, and the action of gravity [12].

We found no difference in the number of small (larger than 10,000 $\mu m^2$) flow voids. Since in previous papers, age-related increase of flow voids larger than 10,000 $\mu m^2$ was shown in healthy individuals [10], we suggest that this parameter might be less reliable than evaluation of 25,000 $\mu m^2$ flow voids in the assessment of choriocapillaris perfusion in CSC. Indeed, if age-related spontaneous occurrence of flow void in choriocapillaris is random, in healthy eye there is a reduced probability of the coincidence of several small voids. Alternatively, if the occurrence of flow voids relates to an abnormal condition, the likelihood for coincidence of several small voids increases, especially if there is a specific topical association between choriocapillaris hypoperfusion and choroidal structures, for example pachyvessels. In such a case, several neighboring small voids appear as a large one.

In recent studies, the evaluation of flow voids in the choriocapillaris slab is one of the most widely used methods for the evaluation of choriocapillaris perfusion. Although the reliability and reproducibility of this approach does not raise any concern, additional software and specific image processing are required. In our study, the percentage of the flow area demonstrated similar results and appeared to be a more straightforward approach.

This study has several limitations, including its retrospective design and the relatively small study population. Another limitation was the use of spectral-domain OCT. Since swept-source OCT operates with a longer wavelength, it penetrated deeper behind the RPE. Although swept source-OCT is potentially less sensitive to structural RPE changes, it does not allow the complete avoidance of RPE and neurosensory retina-related artifacts [13]. In a similar study, therefore, the use of swept source-OCT would lead to numerically different results but would be unlikely to alter significantly the main conclusions of such a study. Another limitation is the inclusion in the study of both acute and resolved CSC since changes of choriocapillaris perfusion may occur after resolution of acute disease. However, only two cases with SRF were included and none of them was a classical center-involving CSC, therefore, the study cohort is relatively homogenous and mostly includes resolved cases. Additionally, some concern might be raised of the potential effect of axial scaling on the assessment of the flow voids. Although we did not apply scaling correction in every case, the mean axial length in our study population was close to emmetropia (23.28 ± 0.78 mm, varying from 21.63 to 24.91 mm). This allowed us to consider the potential effect of magnification as relatively small.

In conclusion, in this study, we confirmed the presence of the general decrease of choriocapillaris perfusion in both eyes of CSC patients, irrespective of the presence of SRF or asymptomatic structural RPE changes. Choriocapillaris ischemia may, therefore, be considered as one of the basic elements in the alteration of intraocular structures in CSC.

**Author Contributions:** Conceptualization: D.S.M. and M.A.B.; methodology: D.S.M.; software: M.A.B.; validation: A.N.K. and D.S.M.; formal analysis: D.S.M.; investigation: D.S.M. and M.A.B.; resources: A.N.K.; data curation: D.S.M.; writing—original draft preparation: M.A.B.; writing—review and editing: D.S.M. and A.N.K.; visualization: M.A.B.; supervision: A.N.K.; project administration: D.S.M.; funding acquisition: NA. All authors have read and agreed to the published version of the manuscript.

**Funding:** This research received no external funding.

**Institutional Review Board Statement:** The study was conducted according to the guidelines of the Declaration of Helsinki, and approved by the Local Ethics Committee of Military Medical Academy (extract from protocol #232, 18/02/2020).

**Conflicts of Interest:** The authors declare no conflict of interest.

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
