# Peer review of "Artifact-Free Evaluation of Choriocapillaris Perfusion in Central Serous Chorioretinopathy"

_2411-5150, 2020_

Round 1

Reviewer 1 Report

1- Figure 1: segmented (yellow) area on c appears significantly larger than the artefact area in A. What was the authors’ definition of artefact segmentation/delineation?

2- Can authors provide more representative images? For instance, scans of different artefacts and representative OCTA choriocapillaris images from healthy/CSC/fellow eyes.

3- Authors assigned cut-off values for flow voids of 10000 and 25000 um2. What is the source of these values? Are they arbitrary values or based on the normal size of intercapillary space in choriocapillaris? Please clarify and provide reference if needed.

4- In methods (lines 90-91): authors assumed that each pixel covers an area of 25 um2, how did authors come up with this value? Authors used 3x3 mm scans consisting of 304x304 pixels. By doing the maths, each pixel should represent an area of about 97.4 um2, not 25. These measurements are based on the assumption that scans cover an exact area of 3x3 mm on the retina, which is not always the case due to the variation in axial length between eyes. Authors might need to mention whether their measurements were corrected for the effect of optical magnification. If correction was not performed, they might need to add it to limitations.

5- What’s the default definition of choriocapillaris slab on the system?

6- Can authors change the order of columns in table 1 so that measurements from different groups are reported first, then p values?

7- Did authors use paired t test to compare between both eyes of CSC patients?

8- Minor comments:

- Line 56: … of any know known comorbidity …

- Lind 66: authors might need to avoid describing the fellow eye of CSC cases as “healthy”.

Author Response

Dear Reviewer,

We would like to thank you for your time and effort in considering the manuscript for publication. Please find below our responses to your comments and general tracking of changes.

1

Your comment

Figure 1: segmented (yellow) area on c appears significantly larger than the artefact area in A. What was the authors’ definition of artefact segmentation/delineation?

Our response

The procedure for defining the borders for RPE and NSS alteration is described in lines 76-79. In the revised manuscript we have clarified this point as follows: “The borders of these abnormal regions were confirmed by the revision of B-scans. To avoid possible inclusion of some abnormal borderline areas, only outer borders of abnormal regions have been delineated.”

In our study, it was important to exclude all abnormal areas of RPE and neurosensory retina from the analysis as potential causes of flow signal attenuation. We, therefore, outlined the abnormal areas along the outer borders of each lesion to avoid a possible inclusion of some borderline areas of these abnormalities in the analysis. This resulted in some expansion of excluded zones over actual abnormal ones.

Lines related to the changes

80-82

2

Your comment

Can authors provide more representative images? For instance, scans of different artefacts and representative OCTA choriocapillaris images from healthy/CSC/fellow eyes.

Our response

We have added figure 2 with representative images of various artifacts in healthy/CSC/fellow eyes.

Lines related to the changes

Figure 2

3

Your comment

Authors assigned cut-off values for flow voids of 10000 and 25000 um2. What is the source of these values? Are they arbitrary values or based on the normal size of intercapillary space in choriocapillaris? Please clarify and provide reference if needed

Our response

The size of flow voids of 10 000 µm2 for evaluation of choriocapillaris status using Phansalkar method was proposed by Spaide [1]. This was further used in other studies [2]. However, the larger flow void area (40 000 µm2) showed better results in another study [3]. We, therefore, decided to use two cut-off values for our paper, one of which should be higher than “conventional”.

  1. Spaide RF. Choriocapillaris Flow Features Follow a Power Law Distribution: Implications for Characterization and Mechanisms of Disease Progression. Am J Ophthalmol. 2016;170:58-67.
  2. Matet A, Daruich A, Hardy S, Behar-Cohen F. PATTERNS OF CHORIOCAPILLARIS FLOW SIGNAL VOIDS IN CENTRAL SEROUS CHORIORETINOPATHY: An Optical Coherence Tomography Angiography Study. Retina. 2019 Nov;39(11):2178-2188.
  3. Spaide RF. Сhoriocapillaris signal voids in maternally inherited diabetes and deafness and in pseudoxanthoma elasticum. Retina. 2017;37(11):2008-2014.

In the revised manuscript, we have provided the rationale for choosing cut-off values for flow voids as well as corresponding references.

Lines related to the changes

111-114

4

Your comment

In methods (lines 90-91): authors assumed that each pixel covers an area of 25 um2, how did authors come up with this value? Authors used 3x3 mm scans consisting of 304x304 pixels. By doing the maths, each pixel should represent an area of about 97.4 um2, not 25. These measurements are based on the assumption that scans cover an exact area of 3x3 mm on the retina, which is not always the case due to the variation in axial length between eyes. Authors might need to mention whether their measurements were corrected for the effect of optical magnification. If correction was not performed, they might need to add it to limitations.

Our response

In the original version of the manuscript, we provided the number of A- and B-scans which OCTA pattern consisted of. At the same time, the resultant image used for analysis has a resolution of 600 per 600 pixels. One pixel, therefore, occupies 25 µm2

All patients included in this study had axial length measurements. The mean value of AL was close to emmetropia 23.28 ± 0.78 mm (varied from 21.63 to 24.91) which allows us to consider the potential effect of magnification as relatively small. However, exact magnification was not taken into account in every case, and we described this fact in the limitations section.

Lines related to the changes

67-68

217-221

5

Your comment

What’s the default definition of choriocapillaris slab on the system?

Our response

The standard slab was generated between two Bruch’s membrane segmentation lines with -9 and -31 µm offset.

We have provided the settings of the standard choriocapillaris slab in the revised version of the manuscript.

Lines related to the changes

76

6

Your comment

Can authors change the order of columns in table 1 so that measurements from different groups are reported first, then p values?

Our response

We have modified table 1 accordingly.

Lines related to the changes

Table 1

7

Your comment

Did authors use paired t test to compare between both eyes of CSC patients?

Our response

Yes, we used the paired t-test to compare measurements between eyes of CSC patients. Thank you for this note. We have pointed this out in the revised manuscript

Lines related to the changes

117

8

Your comment

- Line 56: … of any know known comorbidity …

Our response

Thank you, we have corrected this typo.

Lines related to the changes

60

9

Your comment

- Lind 66: authors might need to avoid describing the fellow eye of CSC cases as “healthy”.

Our response

We agree with the reviewer. We have removed “healthy” from this phrase.

Lines related to the changes

70

Reviewer 2 Report

Review : Artefacts free evaluation of CC in CSCR (Vision)

  1. Introduction and discussion: please refer to the paper : Cakir B, Reich M, Lang S, Bühler A, Ehlken C, Grundel B, Stech M, Reichl S, Stahl A, Böhringer D, Agostini H, Lange C. OCT Angiography of the Choriocapillaris in Central Serous Chorioretinopathy: A Quantitative Subgroup Analysis. Ophthalmol Ther. 2019 Mar;8(1):75-86. doi: 10.1007/s40123-018-0159-1. Epub 2019 Jan 7. PMID: 30617944; PMCID: PMC6393260.; RPE alterations are taken into account in the analysis of CC
  2. Authors analyze acute and resolved cases in the same cohort. In my opinion separate analyses of these subgroups is necessary as choroidal flow might change in the course of the disease.
  3. Authors should take into consideration, that eliminating of areas of SRF presence in the CC flow analysis influences the final conclusions. As it is a safe method, to evaluate the CC flow in relatively unaffected portion of the choroid, still has to have impact on the final analysis. This should be commented on.
  4. We really do not have precise information on the CC flow under the neurosensory retinal detachment or RPE alterations due to technical reasons. Please comment on that – how this influences the evaluation of the choroidal flow as a whole. Is the analyzed area representative for the whole choroidal flow ?
  5. Table 1 should be reorganized . P values should be placed after each of the comparisons. I think something is wrong with this table. The data do not match the subtitles.
  6. Table 1: subtitle The ratio of the number of flow voids larger than 10000 μm2 to the total area of scan after 118 exclusion of neurosensory retina and RPE alterations in eyes of healthy subjects (7.59 ± 1.37 119 voids/mm2) was not statistically significantly different than that in CSC eyes (7.89 ± 1.33 voids/mm2, 120 Ń€ = 0.27) or fellow eyes of CSC patients (8.10 ± 1.20 voids/mm2, p = 0.51). It is not consistent with the data from the table
  7. Table 1. Subtitle: The ratio of the number of flow voids larger than 25000 μm2 to the total area of scan after 122 exclusion of neurosensory retina and RPE alterations in eyes of healthy subjects (1.32 ± 0.68 123 voids/mm2) was not statistically significantly different than that in CSC eyes (2.31 ± 0.92 voids/mm2, 124 Ń€ = 0.0006) or fellow eyes of CSC patients (1.81 ± 0.75 voids/mm2, p = 0.054). Are you sure that it is correct ? I do not think so. Please amend.
  8. Results presented in the table 1 clearly show that CC flow disturbances if present are localized under the areas excluded from the analysis. Please comment.
  9. Please emphasize what is the most important conclusion from the above : CC flow in the unaffected areas is not disturbed.
  10. Discussion: please elaborate more on the importance and origin of large areas of signal void in CSCR. Please expand discussion on potential differences in that respect between resolved and active cases.

Author Response

Dear Reviewer,

We would like to thank you for your time and effort in considering the manuscript for publication. Please find below our responses to your comments and general tracking of changes.

1

Your comment

Introduction and discussion: please refer to the paper : Cakir B, Reich M, Lang S, Bühler A, Ehlken C, Grundel B, Stech M, Reichl S, Stahl A, Böhringer D, Agostini H, Lange C. OCT Angiography of the Choriocapillaris in Central Serous Chorioretinopathy: A Quantitative Subgroup Analysis. Ophthalmol Ther. 2019 Mar;8(1):75-86. doi: 10.1007/s40123-018-0159-1. Epub 2019 Jan 7. PMID: 30617944; PMCID: PMC6393260.; RPE alterations are taken into account in the analysis of CC.

Our response

Thank you. We have read this paper with great interest. In that paper, OCTA signal was evaluated with respect to the RPE alterations and presence of SRF. The paper showed an increase of flow signal beneath atrophic RPE changes and a decrease of that beneath the areas of SRF accumulation. 

In the revised version of the manuscript, we have discussed this paper in the introduction section.

Lines related to the changes

47-51

2

Your comment

Authors analyze acute and resolved cases in the same cohort. In my opinion separate analyses of these subgroups is necessary as choroidal flow might change in the course of the disease.

Our response

Although acute CSC was included in the study, no cases with classical fovea-involved CSC were among them. Moreover, only 2 cases had SRF. Therefore, the study population was relatively homogenous and consisted mostly of resolved cases.

We have clarified this point in the material and methods section and included it in the limitations section.

Lines related to the changes

126

213-217

3

Your comment

Authors should take into consideration, that eliminating of areas of SRF presence in the CC flow analysis influences the final conclusions. As it is a safe method, to evaluate the CC flow in relatively unaffected portion of the choroid, still has to have impact on the final analysis. This should be commented on.

Our response

We took account of differences in the size of excluded altered zones between cases. For all areas excluded, the total area was calculated, and the final value (flow voids number) was calculated per mm2 not per image as a whole. This allowed us to obtain comparable results for cases with different areas of alteration.

In the original paper this point was explained in the material and methods section.

Lines related to the changes

110-111

4

Your comment

We really do not have precise information on the CC flow under the neurosensory retinal detachment or RPE alterations due to technical reasons. Please comment on that – how this influences the evaluation of the choroidal flow as a whole. Is the analyzed area representative for the whole choroidal flow ?

Our response

We agree with the reviewer. There are no papers which reliably show preservation or loss of microcirculation in choriocapillaris under RPE alterations.  We suggest that neurosensory retinal detachment is unlikely to be related to the actual changes in choriocapillaris flow since the distribution of SRF depends on tertiary factors (location of the leak, the volume of the fluid, and action of gravity, etc.). The paper involving SS-OCT partially confirms this.

[Reich M, Boehringer D, Rothaus K, Cakir B, Bucher F, Daniel M, Lang SJ, Lagrèze WA, Agostini H, Lange C. Swept-source optical coherence tomography angiography alleviates shadowing artifacts caused by subretinal fluid. Int Ophthalmol. 2020;40:2007-2016.]

However, this might not be true for RPE alterations since RPE and choriocapillaris are very closely related.

In the revised version of the manuscript, we have considered this point in the introduction as well as in the discussion section.

Lines related to the changes

187-191

5

Your comment

Table 1 should be reorganized . P values should be placed after each of the comparisons. I think something is wrong with this table. The data do not match the subtitles.

Our response

We have modified table 1.

Lines related to the changes

Table 1

6

Your comment

Table 1: subtitle The ratio of the number of flow voids larger than 10000 μm2 to the total area of scan after 118 exclusion of neurosensory retina and RPE alterations in eyes of healthy subjects (7.59 ± 1.37 119 voids/mm2) was not statistically significantly different than that in CSC eyes (7.89 ± 1.33 voids/mm2, 120 Ń€ = 0.27) or fellow eyes of CSC patients (8.10 ± 1.20 voids/mm2, p = 0.51). It is not consistent with the data from the table

Our response

All the data in the text are presented in the table 1. Unfortunately, p-values in the text were swapped.

We have corrected this point in the revised manuscript.

Lines related to the changes

139, 143, 144

7

Your comment

Subtitle: The ratio of the number of flow voids larger than 25000 μm2 to the total area of scan after 122 exclusion of neurosensory retina and RPE alterations in eyes of healthy subjects (1.32 ± 0.68 123 voids/mm2) was not statistically significantly different than that in CSC eyes (2.31 ± 0.92 voids/mm2, 124 Ń€ = 0.0006) or fellow eyes of CSC patients (1.81 ± 0.75 voids/mm2, p = 0.054). Are you sure that it is correct ? I do not think so. Please amend.

Our response

P-values in the text were swapped.

We have corrected this point in the revised manuscript.

Lines related to the changes

139, 143, 144

8

Your comment

Results presented in the table 1 clearly show that CC flow disturbances if present are localized under the areas excluded from the analysis. Please comment.

Our response

We agree with the reviewer. The reduced flow signal beneath affected areas became apparent, but this is not the main result of the study because we do not know the true status of choriocapillaris perfusion under these areas (at least under RPE alterations).  This latter fact was the reason for choosing as the primary aim of the study an assessment of the choriocapillaris status outside altered regions. This point was discussed in lines 173-175.

However, the data leads us to the conclusion that the difference between acute and fellow eye of CSC patients is mainly caused by the higher prevalence of RPE abnormalities in manifest CSC (186-187).

Lines related to the changes

NA

9

Your comment

Please emphasize what is the most important conclusion from the above : CC flow in the unaffected areas is not disturbed.

Our response

Since the eyes of CSC patients are different compared to healthy eyes even after excluding all sources of flow artifacts, we have concluded that there is a general (not regional) reduction of choriocapillaris perfusion in those eyes. This is discussed in the conclusion section.

This point was discussed in lines 182-184.

Lines related to the changes

NA

10

Your comment

Discussion: please elaborate more on the importance and origin of large areas of signal void in CSCR. Please expand discussion on potential differences in that respect between resolved and active cases

Our response

In the revised manuscript, we have discussed the importance and origin of large areas of flow void in CSCR. Potential differences of choriocapillaris perfusion between resolved and active cases were discussed in the limitations section.

Lines related to the changes

196-201

213-217

Your comment

English language and style

(x) Moderate English changes required

Our response

We have paid additional attention to the grammatical and punctuation errors throughout the manuscript. Additionally, the native speaker carefully revised the manuscript.

Round 2

Reviewer 2 Report

Thank you for all the amendments . 

Good job !